# Towards Resource-Efficient Streaming of Large-Scale Medical Image Datasets for Deep Learning

**Pranav Kulkarni**[1,2] (iD)                                    PKULKARNI@SOM.UMARYLAND.EDU

**Adway Kanhere**[1,2] (iD)                                    AKANHERE@SOM.UMARYLAND.EDU

**Eliot L. Siegel**[1]                                          ESIEGEL@SOM.UMARYLAND.EDU

**Paul H. Yi**[3] (iD)                                                PAUL.YI@STJUDE.ORG

**Vishwa S. Parekh**[4,5] (iD)                              VISHWA.S.PAREKH@UTH.TMC.EDU

[1] *University of Maryland School of Medicine, Baltimore, MD*

[2] *University of Maryland Institute for Health Computing, North Bethesda, MD*

[3] *St. Jude Children's Research Hospital, Memphis, TN*

[4] *UTHealth Houston, Houston, TX*

[5] *Johns Hopkins University School of Medicine, Baltimore, MD*

**Editors:** Accepted for publication at MIDL 2025

## Abstract

Large-scale medical imaging datasets have accelerated deep learning (DL) for medical image analysis. However, the large scale of these datasets poses a challenge for researchers, resulting in increased storage and bandwidth requirements for hosting and accessing them. Since different researchers have different use cases and require different resolutions or formats for DL, it is neither feasible to anticipate every researcher's needs nor practical to store data in multiple resolutions and formats. To that end, we propose the Medical Image Streaming Toolkit (MIST), a format-agnostic database that enables streaming of medical images at different resolutions and formats from a single high-resolution copy. We evaluated MIST across eight popular, large-scale medical imaging datasets spanning different body parts, modalities, and formats. Our results showed that our framework reduced the storage and bandwidth requirements for hosting and downloading datasets without impacting image quality. We demonstrate that MIST addresses the challenges posed by large-scale medical imaging datasets by building a data-efficient and format-agnostic database to meet the diverse needs of researchers and reduce barriers to DL research in medical imaging.

**Keywords:** Imaging Databases, Compression, Progressive Streaming, Data-Efficiency

## 1. Introduction

The curation of large-scale, publicly accessible medical imaging datasets has accelerated the development of deep learning (DL) models for medical image analysis (Hosny et al., 2018). However, the large scale of medical imaging datasets poses a significant challenge for researchers both hosting or accessing them (Jia et al., 2023; Li et al., 2023; Magudia et al., 2021). For researchers hosting these datasets, the growing collection of curated datasets and increasing demand for access result in greater costs for managing storage and networking infrastructures (Doo et al., 2024). On the other hand, researchers accessing the datasets require sufficient compute and storage resources to download and process hundreds of thousands of high-resolution medical images to be "DL-ready", especially as datasets continue to grow in scale and resolution (Jia et al., 2023; Kiryati and Landau, 2021).

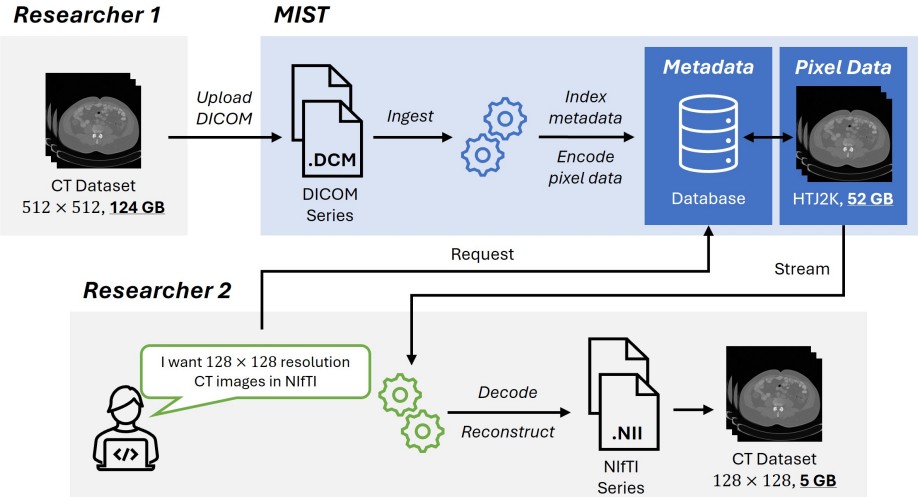

Figure 1: An overview of MIST. Suppose a researcher uploads a high-resolution DICOM CT dataset to MIST. It is ingested and stored in a format-agnostic representation, where the pixel data is progressively encoded and separated from its associated metadata. This allows another researcher to access the dataset at a lower resolution in NIfTI format without data duplication.

Accessing and utilizing large-scale medical imaging datasets for DL research presents several challenges, particularly in terms of data storage, bandwidth limitations, and pre-processing requirements. For instance, training a DL model for early-stage lung cancer detection using the NLST dataset (Team, 2011) would require downloading over 11 TB of CT scans, which presents substantial storage challenges for many research environments. Furthermore, limited bandwidth can result in download times spanning several days, creating significant logistical bottlenecks. Beyond storage and bandwidth constraints, researchers often require data in different formats depending on their specific tasks. For example, 3D segmentation models necessitate preprocessing raw DICOM series into NIfTI format (Li et al., 2016), while CNN-based classification models may require lower-resolution inputs, such as 224×224 images. Unfortunately, this inefficiency scales with the number of researchers accessing the dataset. Given the diversity of research needs, it is neither feasible to anticipate all possible data configurations nor practical for each researcher to store, pre-process, and manage such large-scale datasets independently. These inefficiencies highlight the need for scalable, adaptive data access solutions that can accommodate varying research requirements while minimizing redundant computational and storage overhead.

To that end, we propose the Medical Image Streaming Toolkit (MIST), a format-agnostic imaging database that enables streaming of medical images at different resolutions and formats from a single high-resolution copy (Figure 1). Our goal is to address the challenges posed by large-scale medical imaging datasets by building a data-efficient and format-agnostic database to meet the diverse needs of researchers in the medical imaging with DL community. The purpose of this study is to evaluate MIST for reducing the storage

and bandwidth requirements for hosting and downloading eight large-scale medical imaging datasets spanning different body parts, modalities, and formats, without impacting image quality.

## 2. Background

One core challenge is the lack of interoperability of DICOM with popular DL frameworks (e.g., PyTorch) due to its complex multi-file structure and varying metadata tags, necessitating extensive domain expertise to preprocess DICOM series prior to training DL models (Galbusera and Cina, 2024; Botnari et al., 2024). As a result, "DL-ready" formats (e.g., NIfTI) that are easier to use with DL frameworks have become popular among researchers. However, datasets curated as part of clinical studies with applications beyond DL are often released as DICOM databases, due to the widespread use of DICOM in clinical settings and its rich, clinically relevant metadata. The difference between formats, based on their clinical relevance and ease of use, has created a distinct format hierarchy. As a result, some collections, such as UPENN-GBM (Bakas et al., 2022), have released both DICOM and NIfTI versions of the dataset to meet the diverse needs of researchers using them. However, this results in data duplication by hosting the same dataset in multiple different formats, requiring additional storage to do so (Jiménez-Sánchez et al., 2024).

Another challenge is that many researchers may not need high-resolution medical images for developing DL models, as images are often downsampled to lower resolutions to reduce computational costs and potentially improve generalizability to unseen data (Sabottke and Spieler, 2020). Consider MIMIC-CXR, a dataset containing 4.7 TB of chest x-rays (CXRs) with mean resolution $2500 \times 3056$ (Johnson et al., 2019). If a researcher wanted to use this dataset for training a CNN-based classifier, they would have to downsample the CXRs by more than 100 times to $224 \times 224$. This approach is highly inefficient because not only does this necessitate downloading and storing the full 4.7 TB dataset, but also requires considerable computational resources to downsample the CXRs.

In our prior work, we explored solutions to both challenges using progressive encoding in order to accelerate DL inference in clinical deployment, where latency is critical (Kulkarni et al., 2024). Here, we primarily focus on building a flexible and scalable framework for hosting and sharing medical imaging datasets in research settings, with a focus on reducing data storage and transmission. Our approach is not intended for real-time DL applications, but rather for efficient data access across diverse research settings, where datasets are typically downloaded before being used for DL tasks.

## 3. Methods

### 3.1. Medical Image Streaming Toolkit

MIST is a format-agnostic database for hosting large-scale medical imaging datasets for developing DL models. Our framework allows a researcher to stream medical images at different resolutions and formats from a single high-resolution copy. Compared to existing solutions, MIST differs in three key aspects (Figure 2):

**Format Hierarchy:** At the core of MIST is its format-agnostic design. We ingested and stored medical images in a format-agnostic representation, separating the pixel data from

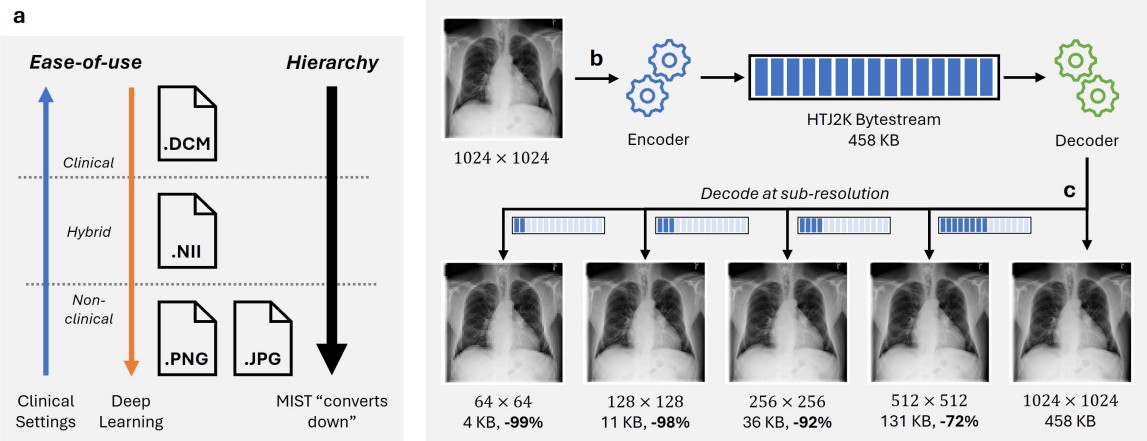

Figure 2: **(a)** An illustration of MIST's format hierarchy, where medical images are always "converted down" to other formats. **(b)** Medical images are compressed by progressively encoding pixel data into a bytestream using HTJ2K. **(c)** HTJ2K allows sub-resolution images to be accessed by decoding a partial bytestream.

associated metadata (Figure 1). Similar approaches have been used by AWS HealthImaging[1] and the NCI Imaging Data Commons (Fedorov et al., 2021) for hosting medical image datasets in DICOM. Our method differs by allowing medical images in DICOM, NIfTI, or other non-medical imaging formats (e.g., JPEG, PNG) to be ingested and stored in the same format-agnostic representation.

The format-agnostic representation allows medical images to be "converted down" into other formats using the pixel data and associated metadata when streaming to researchers. To achieve this, we followed a strict format hierarchy, based on the availability of metadata, to govern how a medical image in one format is streaming in a different format (Figure 2a). Specifically, a DICOM series can be streamed in DICOM, NIfTI, or other non-medical imaging formats, whereas a NIfTI series can only be streamed in NIfTI or other non-medical imaging formats. This limitation is because NIfTI lacks the metadata to "convert up" to DICOM, while DICOM can "convert down" to NIfTI. Similarly, non-medical imaging formats can only be streamed in other non-medical imaging formats.

When a medical image is encoded by MIST, we extracted and stored its associated metadata tags in a separate JSON file. Format conversion is handled by mapping the relevant metadata tags from the source format to equivalent tags in the target format (e.g., pixel spacing when converting from DICOM to NIfTI). This enables metadata to be preserved when converting to a different format, even when the target format is a non-medical imaging one (e.g., PNG).

**Medical Image Compression:** Medical image compression is used to reduce the file size of medical images, thereby requiring less resources to store and transmit large-scale medical image datasets (Koff and Shulman, 2006). It works by encoding pixel data into a bytestream that can be decoded to reconstruct the original pixel data. We used the

---

1. https://aws.amazon.com/healthimaging

High-Throughput JPEG 2000 (HTJ2K) codec to encode pixel data (Figure 2b). HTJ2K is a progressive encoding format that encodes pixel data as a series of lossy decompositions, with each subsequent one having a higher resolution (Taubman et al., 2019, 2020). When fully decoded, the pixel data is reconstructed without any data loss.

We used OpenJPHpy[2] to encode pixel data as lossless with 16-bit depth, $64 \times 64$ block size, $n$ decomposition levels, tile-part divisions by resolution, and tile-part markers to identify the location of decompositions within the bytestream. The number of decomposition levels $n$ was calculated for each medical image by (Kulkarni et al., 2024):

$$n = \left\lfloor \log_2 \frac{\min(M, N)}{\alpha} \right\rfloor \tag{1}$$

where $M \times N$ was the pixel data resolution and $\alpha = 64$ was an arbitrarily set lower bound such that the first decomposition will not reconstruct an image smaller than $64 \times 64$. For a DICOM series and other non-medical imaging formats (e.g., JPEG, PNG), each image (e.g., slice in a volume) was encoded and stored separately. On the other hand, NIfTI stores the entire volume in a single file. To maintain consistency, we encoded and stored each slice of the NIfTI volume separately. We computed rescaling intercept and slope before encoding the pixel data because existing HTJ2K implementations support up to uint16 pixel data.

**Access to Sub-Resolution Images:** An advantage of progressive encoding is the ability to access sub-resolution images from partial bytestreams of a single high-resolution copy (Foos et al., 2000; Noumeir and Pambrun, 2011). Pixel data encoded using HTJ2K with $n$ decompositions levels contains $n + 1$ decompositions, where $i$th decomposition has a resolution that is $1/2^{n+1-i}$ of the original resolution (Figure 2c). For example, an abdominal CT scan of resolution $512 \times 512$, encoded with $n = 3$ decomposition levels, can be decoded at sub-resolutions $64 \times 64$, $128 \times 128$, $256 \times 256$, and $512 \times 512$.

If the pixel data is encoded with tile-part divisions by resolution and the bytes required to reconstruct the $i$th decomposition are available, the image can be decoded at decomposition level $i - 1$ and all previous levels without needing the complete bytestream (Kulkarni et al., 2024). This allows researchers to download sub-resolution images using a partial bytestream. For example, a CXR with resolution $1024 \times 1024$ can be streamed at resolution $256 \times 256$ using the first 35 KB of the complete 448 KB bytestream.

To achieve this, we created a new entry to each medical image's metadata that mapped decompositions to the location of tile-part markers in the bytestream. The tile-part markers define the extent of the bytestream required to decode a specific decomposition. They were identified by the start of tile-part marker (bytes 0xFF90) and end of codestream marker (bytes 0xFFD9) (Boliek et al., 2000). Once the pixel data is decoded at a sub-resolution, the metadata for the affine transformation matrix and voxel size is rescaled to ensure accurate transformation to world coordinates.

### 3.2. Experiments

In this study, we evaluated MIST for reducing the storage and bandwidth requirements for hosting and downloading using eight popular, large-scale medical imaging datasets for developing DL models, spanning different body parts, modalities, and formats (Table 1).

---

2. https://github.com/BioIntelligence-Lab/openjphpy

Table 1: Summary of datasets.

| Dataset | Modality | Body Part | # Series | Format | Size (GB) |
|---|---|---|---|---|---|
| NIH-ChestX-ray14 (Wang et al., 2017) | CR | Chest | 112,120 | PNG | 41.96 |
| MSD-Liver (Antonelli et al., 2022) | CT | Abdomen | 332 | NIfTI | 26.94 |
| AMOS (Ji et al., 2022) | CT, MR | Abdomen | 960 | NIfTI | 22.59 |
| LIDC-IDRI (Armato III et al., 2011) | CT | Chest | 1,308 | DICOM | 124.00 |
| NSCLC-Radiomics (Aerts et al., 2014) | CT | Lung | 422 | DICOM | 33.32 |
| TCGA-KIRC (Shinagare et al., 2015) | CT, MR | Kidney | 2,654 | DICOM | 85.28 |
| TCGA-BRCA (Guo et al., 2015) | MG, MR | Breast | 1,877 | DICOM | 82.08 |
| UPENN-GBM (Bakas et al., 2022) | MR | Brain | 3,680 | DICOM | 129.83 |

Detailed descriptions of each dataset is provided in Appendix A. Our code is available at: https://github.com/BioIntelligence-Lab/MIST.

We encoded every medical image in a dataset using HTJ2K and stored them in the format-agnostic representation. Any series that failed to encode was excluded from our analysis. A series may fail due to a lack of pixel data (e.g., SEG DICOM files) or unsupported pixel data (e.g., exceeding 16-bit depth, floating point data). Then, we measured the amount of data stored and transmitted by hosts. We also measured the amount of data transmitted across all decompositions. Finally, we evaluated image quality by comparing all decompositions with the pixel data using the structural similarity index measure (SSIM) and peak signal-to-noise ratio (PSNR). These quantitative metrics were measured by downsampling the pixel data to the sub-resolution (using bilinear interpolation) and rescaling pixel values to range 0-1 using scikit-image. The SSIM between pixel data $X$ and sub-resolution image $Y$ across windows $x$ and $y$ is defined as (Wang et al., 2004):

$$\text{SSIM}(x,y) = \frac{(2\mu_x\mu_y + k_1^2)(2\sigma_{xy} + k_2^2)}{(\mu_x^2 + \mu_y^2 + k1^2)(\sigma_x^2 + \sigma_y^2 + k_2^2)} \tag{2}$$

where $(\mu_x, \sigma_x^2)$ and $(\mu_y, \sigma_y^2)$ are statistics of pixel values in windows $x$ and $y$ respectively, $\sigma_{xy}$ is covariance of pixel values in windows $x$ and $y$, $k_1 = 0.01$, and $k_2 = 0.03$. The PSNR between pixel data $X$ and sub-resolution image $Y$ of fixed resolution $M \times N$ is defined as:

$$\text{PSNR}(X,Y) = 10\log_{10}\left(\frac{MN}{\sum_{m=1}^{M}\sum_{n=1}^{N}(X_{m,n} - Y_{m,n})^2}\right) \tag{3}$$

where $X_{m,n}$ and $Y_{m,n}$ are pixel values of $X$ and $Y$ at index $(m,n)$ respectively. If $X$ and $Y$ are equal (in the case of lossless reconstruction), then PSNR is infinity. As a result, the mean PSNR for a decomposition only considers the non-infinite PSNR values.

## 4. Results

**Medical Image Compression:** We observed that MIST successfully encoded all eight datasets, with the exception of $N = 28$ (2.92%) from AMOS, $N = 7$ (0.26%) series from TCGA-KIRC, and $N = 20$ (1.07%) series from TCGA-BRCA. Excluding these series from our analysis, our method reduced the total amount of data stored and transmitted by hosts from 536.68 GB to 234.77 GB, a decrease of 56.22% (Figure 3a). We observed that

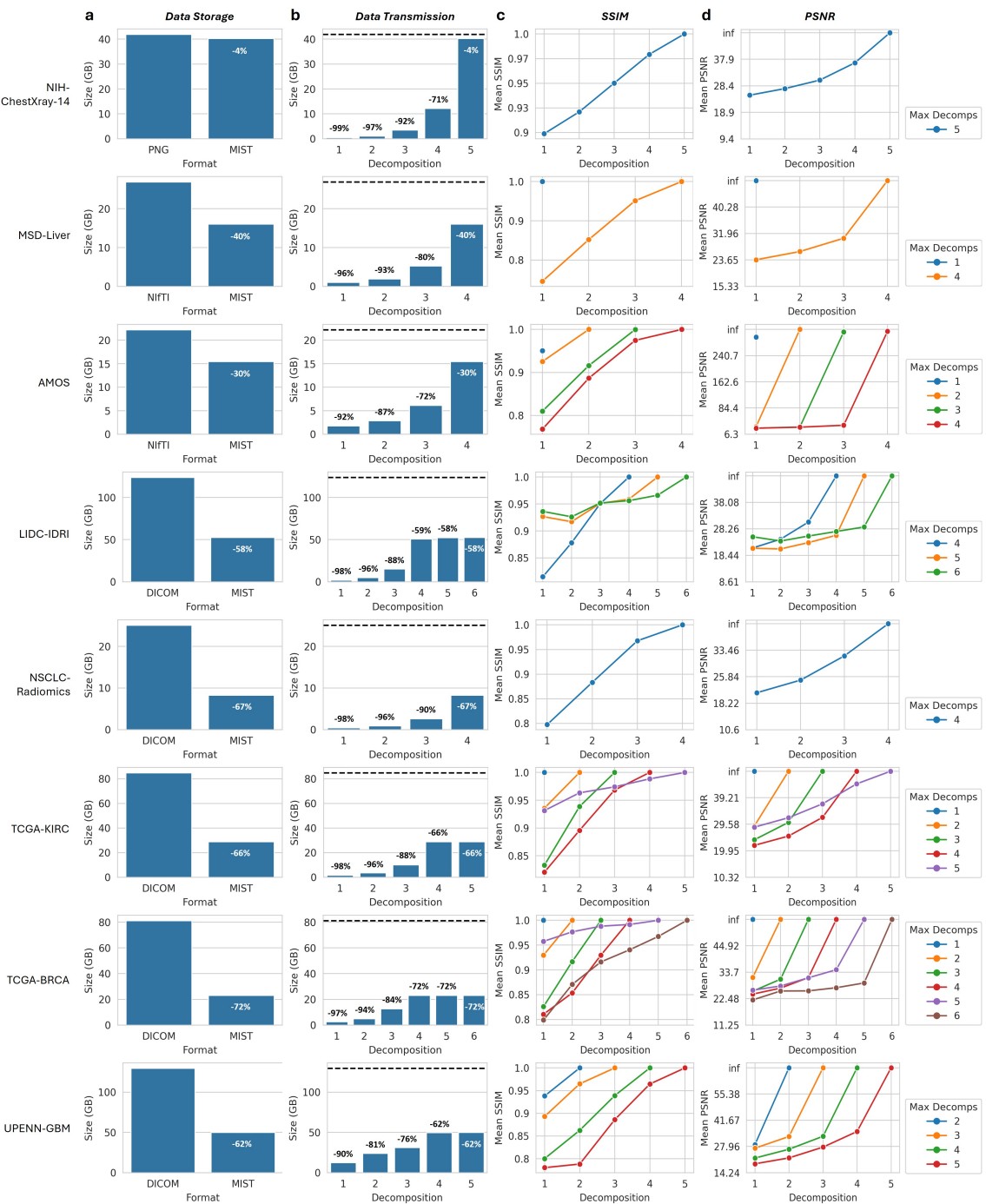

Figure 3: **(a)** Decrease in data storage due to medical image compression. **(b)** Decrease in data transmission due to sub-resolution image access. Image quality measured using **(c)** mean SSIM and **(d)** mean PSNR values across all decompositions, stratified by the maximum decomposition for an image.

TCGA-BRCA had the greatest decrease, from 81.30 GB to 23.13 GB (71.55%), while NIH-ChestX-ray14 had the least decrease, from 41.96 GB to 40.29 GB (4%). On average, MIST reduced data storage by $49.79 \pm 23.27\%$. Detailed results are provided in Table B.1.

**Access to Sub-Resolution Images:** We observed that downloading datasets at sub-resolutions led to a further decrease in the total amount of data transmitted (Figure 3b). At the smallest sub-resolution (i.e., decomposition level 1), MIST reduced total data transmission to 22.59 GB (95.79%), ranging from 98.93% for NIH-ChestX-ray14 to 90.36% for UPENN-GBM, with an average reduction of $96.10 \pm 3.20\%$. At the largest sub-resolution (varies across datasets; see Table B.1), our method reduced total data transmission to 189.76 GB (64.62%), ranging from 89.60% for NSCLC-Radiomics to 30.41% for AMOS, with an average reduction of $66.04 \pm 17.63\%$. Detailed results are provided in Appendix B.

**Evaluation of Image Quality:** We observed that MIST did not impact the image quality of all eight datasets encoded in our study, with a mean SSIM of $1.00 \pm 0.00$ and mean PSNR of infinity. When accessing sub-resolution images, our results show that each subsequent decomposition results in increased image quality as indicated with increasing mean SIIM and PSNR values (Figure 3c–d). At the smallest sub-resolution, we measured a mean SSIM of $0.83 \pm 0.06$, ranging from $0.75 \pm 0.17$ for MSD-Liver to $0.93 \pm 0.04$ for UPENN-GBM. We measured a mean PSNR of $23.77 \pm 2.11$, ranging from $21.18 \pm 1.96$ for LIDC-IDRI to $27.22 \pm 3.02$ for UPENN-GBM.

In addition to quantitative metrics of image quality, we performed a preliminary set of DL experiments across 2D classification and 3D segmentation tasks (Appendix C). Examples of sub-resolution images from each dataset are provided in Figure D.1.

## 5. Discussion

We demonstrated that MIST can reduce the storage and bandwidth requirements for hosting and downloading large-scale medical imaging datasets across different body parts, modalities, and formats. While the decrease in data storage and transmission varies from one dataset to another, our results consistently showed a significant improvement in data-efficiency despite differences in pixel bit depth, resolution, and formats. We observed that MIST-encoded DICOM datasets had the greatest decrease in data storage followed by NIfTI and PNG. This is primarily for two reasons: 1) The DICOM datasets in our study stored uncompressed pixel data, resulting in a high compression ratio. 2) The NIfTI and PNG formats use lossless compression to encode pixel data. However, MIST is able to exceed the compression ratio of NIfTI while closely matching that of PNG (Elhadad et al., 2024).

Our method goes a step further by also allowing streaming of medical images at different resolutions and formats from a single format-agnostic, high-resolution copy. Since every researcher has different needs and use cases, eliminating the preprocessing steps of image resizing and format conversion (e.g., DICOM to NIfTI) before training DL models can lead to greater resource-efficiency (Jia et al., 2023). Moreover, streaming sub-resolution images inherently leads to reduced data transmission to download medical imaging datasets. For example, if a researcher wanted to train a DL model using transfer learning (at resolution $224 \times 224$) with the NIH-ChestX-ray14 dataset, they would have to download 92% less data compared to current solutions.

Despite decreasing the amount of data stored and transmitted, we demonstrated that MIST-encoded datasets preserved image quality, indicated by a lossless reconstruction of pixel data, across different body parts, modalities, and formats. However, accessing sub-resolution images led to a decrease in image quality due to the lossy reconstruction of pixel data from partial bytestreams. When comparing with the original pixel data, differences in image quality metrics were due to variations in resizing methods, such as compression artifacts from progressive decoding vs bilinear interpolation in conventional downsampling. However, this does not inherently indicate a loss of image quality, but rather reflects the differences in resizing techniques (Botnari et al., 2024). In Appendix C and our prior work, we showed that sub-resolution images did not impact the performance of DL models across 2D classification and 3D segmentation tasks (Kulkarni et al., 2024).

There are some limitations of our work: 1) While we observe no impact on image quality using quantitive metrics, such as SSIM and PSNR, there is a need to further validate MIST-encoded datasets for training DL models. 2) Since we measured data-efficiency metrics for the entire dataset, where each image can have a different resolution and, thus, number of decomposition levels, the decrease in data storage and transmission may dramatically vary from one image to another. 3) Due to limitations in current HTJ2K implementation, we excluded series that contained unsupported pixel data. For example, $n = 28$ MRI volumes from AMOS exceeded the 16-bit pixel depth, potentially due to the presence of artifacts, and were excluded. 4) For NIfTI volumes, each slice is encoded and decoded as a separate image, adding an additional step to reconstruct the volume after decoding. 5) Further evaluation is required to identify the best approach of handling complex DICOM metadata. In our experiments, we used JSON to store metadata, but this may not scale efficiently when the metadata is complex. For future work, we intend to address these limitations and plan to extensively validate MIST in real-world research settings.

In conclusion, MIST is a crucial first step towards addressing the challenges posed by hosting and accessing large-scale medical imaging datasets, especially as datasets continue to grow in scale and resolution. Our work demonstrates that a data-efficient, format-agnostic database can not only reduce data storage and transmission, but also meet the diverse needs of researchers and reduce barriers to DL research in medical imaging.

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

## Appendix A. Dataset Descriptions

**NIH-ChestX-ray14** The National Institutes of Health (NIH) ChestX-ray14 dataset contains $n = 112, 120$ frontal CXRs from 30,805 patients (Wang et al., 2017). Each CXR is annotated with the presence of up to 14 disease labels. The dataset is available in the PNG format and spans a total size of 41.96 GB.

**MSD-Liver** The Medical Segmentation Decathlon (MSD) is a collection of benchmark datasets for medical image segmentation, spanning 10 tasks across different body parts and modalities (Antonelli et al., 2022). The MSD-Liver dataset is the third task comprising of a subset of $n = 201$ abdominal portal venous phase CT scans from the 2017 Liver Tumor Segmentation (LiTS) challenge (Bilic et al., 2023). Additionally, voxel-level annotations for liver and liver tumors are provided for $n = 131$ CT scans. The dataset is available in the NIfTI format and spans a total size of 26.94 GB.

**AMOS** The Abdominal Multi-Organ Segmentation (AMOS) dataset contains $n = 600$ abdominal CT and MRI scans (Ji et al., 2022). Additionally, voxel-level annotations for 15 abdominal structures are provided for $n = 360$ scans. The dataset is available in the NIfTI format and spans a total size of 22.59 GB.

**LIDC-IDRI** The Lung Image Database Consortium and Image Database Resource Initiative (LIDC-IDRI) dataset contains $n = 1, 308$ lung CT scans (Armato III et al., 2011). Each scan has corresponding lesion annotations from up to four radiologists. The dataset is available in the DICOM format on TCIA and spans a total size of 124.00 GB.

**NSCLC-Radiomics** The Non-Small Cell Lung Cancer (NSCLC)-Radiomics dataset comprised of $n = 422$ lung CT scans (Aerts et al., 2014). Each CT scans has corresponding segmentations and 440 extracted radiomics features. The dataset is available in the DICOM format on TCIA and spans a total size of 33.32 GB.

**TCGA-KIRC** The Cancer Genome Atlas Kidney Renal Clear Cell Carcinoma (TCGA-KIRC) dataset contains $n = 2, 654$ abdominal CT and MRI scans (Shinagare et al., 2015). The dataset is available in the DICOM format on TCIA and spans a total size of 85.28 GB.

**TCGA-BRCA** The Cancer Genome Atlas Breast Invasive Carcinoma (TCGA-BRCA) dataset contains $n = 1, 877$ MRI scans and mammograms (Guo et al., 2015). Each scan has a corresponding standardized breast imaging report, genomic, and pathology data. The dataset is available in the DICOM format on TCIA and spans a total size of 82.08 GB.

**UPENN-GBM** the University of Pennsylvania Glioblastoma (UPENN-GBM) dataset contains $n = 3, 680$ multi-parametric brain MRIs (Bakas et al., 2022). Each scan has corresponding segmentations, radiomics, and pathology data. The dataset is available in the DICOM format on TCIA and spans a total size of 129.83 GB.

# Appendix B. Detailed Results

Table B.1: Encoding summary of all eight datasets

| Dataset | Max Decomps. | # Series | | | Size (GB) | |
|---|---|---|---|---|---|---|
| | | All | Excluded | Encoded | Original | MIST |
| NIH-ChestX-ray14 | 5 | 112,120 | 0 | 112,120 | 41.96 | 40.29 ($-4.00\%$) |
| MSD-Liver | 4 | 332 | 0 | 332 | 26.94 | 16.07 ($-40.36\%$) |
| AMOS | 4 | 960 | 28 (2.92%) | 932 | 22.23 | 15.47 ($-30.41\%$) |
| LIDC-IDRI | 6 | 1,308 | 0 | 1,308 | 124.00 | 52.70 ($-57.50\%$) |
| NSCLC-Radiomics | 4 | 422 | 0 | 422 | 25.10 | 8.27 ($-67.04\%$) |
| TCGA-KIRC | 5 | 2,654 | 7 (0.26%) | 2,647 | 84.95 | 28.92 ($-65.96\%$) |
| TCGA-BRCA | 6 | 1,877 | 20 (1.07%) | 1,857 | 81.30 | 23.13 ($-71.55\%$) |
| UPENN-GBM | 5 | 3,680 | 0 | 3,680 | 129.83 | 49.93 ($-61.54\%$) |

Table B.2: Mean data-efficiency and image quality metrics for NIH-ChestX-ray14 dataset.

| Format | Decomp. | Size (GB) | SSIM | PSNR |
|---|---|---|---|---|
| PNG | - | 41.96 | - | - |
| MIST | 1 | 0.45 ($-98.93\%$) | $0.90 \pm 0.02$ | $25.06 \pm 2.59$ |
| | 2 | 1.13 ($-97.30\%$) | $0.92 \pm 0.02$ | $27.40 \pm 3.66$ |
| | 3 | 3.53 ($-91.59\%$) | $0.95 \pm 0.01$ | $30.46 \pm 4.56$ |
| | 4 | 12.19 ($-70.96\%$) | $0.98 \pm 0.01$ | $36.64 \pm 4.43$ |
| | 5 | 40.29 ($-4.00\%$) | $1.00 \pm 0.00$ | inf |

Table B.3: Mean data-efficiency and image quality metrics for MSD-Liver dataset.

| Format | Decomp. | Size (GB) | SSIM | PSNR |
|---|---|---|---|---|
| NIfTI | - | 26.94 | - | - |
| MIST | 1 | 1.04 ($-96.14\%$) | $0.75 \pm 0.17$ | $23.69 \pm 3.56$ |
| | 2 | 1.95 ($-92.76\%$) | $0.85 \pm 0.11$ | $26.28 \pm 3.45$ |
| | 3 | 5.27 ($-80.43\%$) | $0.95 \pm 0.04$ | $30.46 \pm 3.84$ |
| | 4 | 16.07 ($-40.36\%$) | $1.00 \pm 0.00$ | inf |

Table B.4: Mean data-efficiency and image quality metrics for AMOS dataset.

| Format | Decomp. | Size (GB) | SSIM | PSNR |
|---|---|---|---|---|
| NIfTI | - | 22.23 | - | - |
| MIST | 1 | 1.79 ($-91.96\%$) | $0.78 \pm 0.15$ | $24.29 \pm 3.05$ |
| | 2 | 2.89 ($-87.00\%$) | $0.90 \pm 0.08$ | $27.13 \pm 2.88$ |
| | 3 | 6.15 ($-72.33\%$) | $0.98 \pm 0.03$ | $32.85 \pm 3.49$ |
| | 4 | 15.47 ($-30.41\%$) | $1.00 \pm 0.00$ | inf |

Table B.5: Mean data-efficiency and image quality metrics for LIDC-IDRI dataset.

| Format | Decomp. | Size (GB) | SSIM | PSNR |
|---|---|---|---|---|
| DICOM | - | 124.00 | - | - |
| MIST | 1 | 2.07 (−98.33%) | 0.82 ± 0.04 | 21.18 ± 1.96 |
| | 2 | 4.93 (−96.03%) | 0.88 ± 0.04 | 24.42 ± 2.32 |
| | 3 | 15.36 (−87.62%) | 0.95 ± 0.03 | 30.76 ± 2.97 |
| | 4 | 50.80 (−59.03%) | 1.00 ± 0.00 | 25.99 ± 4.95 |
| | 5 | 52.46 (−57.70%) | 1.00 ± 0.00 | 28.97 ± 5.95 |
| | 6 | 52.70 (−57.50%) | 1.00 ± 0.00 | inf |

Table B.6: Mean data-efficiency and image quality metrics for NSCLC-Radiomics dataset.

| Format | Decomp. | Size (GB) | SSIM | PSNR |
|---|---|---|---|---|
| DICOM | - | 25.10 | - | - |
| MIST | 1 | 0.44 (−98.25%) | 0.80 ± 0.03 | 21.23 ± 1.68 |
| | 2 | 0.91 (−96.37%) | 0.88 ± 0.02 | 24.86 ± 2.14 |
| | 3 | 2.61 (−89.60%) | 0.97 ± 0.01 | 31.84 ± 2.82 |
| | 4 | 8.27 (−67.04%) | 1.00 ± 0.00 | inf |

Table B.7: Mean data-efficiency and image quality metrics for TCGA-KIRC dataset.

| Format | Decomp. | Size (GB) | SSIM | PSNR |
|---|---|---|---|---|
| DICOM | - | 84.95 | - | - |
| MIST | 1 | 1.72 (−97.97%) | 0.83 ± 0.05 | 22.31 ± 2.53 |
| | 2 | 3.68 (−95.66%) | 0.90 ± 0.04 | 25.95 ± 3.17 |
| | 3 | 10.17 (−88.03%) | 0.97 ± 0.02 | 32.06 ± 3.30 |
| | 4 | 28.91 (−65.97%) | 1.00 ± 0.00 | 44.23 ± 2.44 |
| | 5 | 28.92 (−65.96%) | 1.00 ± 0.00 | inf |

Table B.8: Mean data-efficiency and image quality metrics for TCGA-BRCA dataset.

| Format | Decomp. | Size (GB) | SSIM | PSNR |
|---|---|---|---|---|
| DICOM | - | 81.30 | - | - |
| MIST | 1 | 2.57 (−96.84%) | 0.83 ± 0.08 | 25.17 ± 3.17 |
| | 2 | 4.99 (−93.86%) | 0.90 ± 0.06 | 28.91 ± 3.89 |
| | 3 | 12.78 (−84.28%) | 0.97 ± 0.04 | 31.26 ± 3.79 |
| | 4 | 23.05 (−71.66%) | 1.00 ± 0.00 | 31.73 ± 8.54 |
| | 5 | 23.09 (−71.60%) | 1.00 ± 0.00 | 29.13 ± 6.83 |
| | 6 | 23.13 (−71.55%) | 1.00 ± 0.00 | inf |

Table B.9: Mean data-efficiency and image quality metrics for UPENN-GBM dataset.

| Format | Decomp. | Size (GB) | SSIM | PSNR |
|---|---|---|---|---|
| DICOM | - | 129.83 | - | - |
| MIST | 1 | 12.51 (−90.36%) | 0.93 ± 0.04 | 27.22 ± 3.02 |
| | 2 | 24.12 (−81.42%) | 0.99 ± 0.04 | 28.84 ± 4.01 |
| | 3 | 31.21 (−75.96%) | 1.00 ± 0.02 | 32.96 ± 2.51 |
| | 4 | 49.77 (−61.66%) | 1.00 ± 0.00 | 35.75 ± 2.61 |
| | 5 | 49.93 (−61.54%) | 1.00 ± 0.00 | inf |

## Appendix C. Preliminary Deep Learning Experiments

We conducted two preliminary experiments to validate MIST-encoded sub-resolution images for training DL models. In both experiments, we trained two models at a fixed input resolution using the original and MIST-encoded images, then tested them on the original internal and external datasets respectively. The MIST-encoded images were decoded at the sub-resolution closest to the model's input resolution.

### C.1. 2D Classification

We randomly split the NIH-ChestX-ray14 dataset on the patient-level into train $(70\%, n = 78,075)$, validation $(10\%, n = 11,079)$, and test $(20\%, n = 22,966)$ sets. We used $n = 15,000$ randomly sampled frontal CXRs from the MIMIC-CXR dataset (Johnson et al., 2019) as our external test set. Then, we trained two ImageNet-pretrained DenseNet121 models at resolution $224 \times 224$, one on the original PNG images and the other on the sub-resolution images. Both models were trained for 100 epochs with binary cross-entropy loss, 64 batch size, 5e-5 learning rate, and random augmentations. We measured the performance of both models using the mean area under the receiver operating characteristic curve (AUROC) scores of seven labels: Atelectasis, Cardiomegaly, Consolidation, Edema, Pleural effusion, Pneumonia, and Pneumothorax. On the NIH-ChestX-ray14 test set, we observed that both the PNG-trained and MIST-trained models measured a mean AUROC of $0.84 \pm 0.05$. When evaluated on the external MIMIC-CXR test set, both models similarly measured a mean AUROC of $0.76 \pm 0.07$.

### C.2. 3D Segmentation

We randomly split the MSD-Liver dataset into train $(80\%, n = 106)$ and test $(20\%, n = 25)$ sets. We used $n = 30$ volumes from the BTCV dataset (Landman et al., 2015) as our external test set. Then, we trained two 3D-UNet models at resolution $256 \times 256 \times 128$ and voxel spacing $1.5 \times 1.5 \times 2.0$. Both models were trained using random foreground patches of resolution $128 \times 128 \times 32$ for 500 epochs with Dice loss, 2 batch size, 1e-4 learning rate with cosine annealing scheduler, and random augmentations. We measured the performance of both models using the mean Dice score. On the MSD-Liver test set, we observed that the NIfTI-trained model measured a mean Dice score of $0.95 \pm 0.02$, while the MIST-trained model measured a mean Dice score of $0.94 \pm 0.02$. When evaluated on the external BTCV test set, both models measured a mean Dice score of $0.93 \pm 0.02$.

# Appendix D. Visualization of Sub-Resolution Images

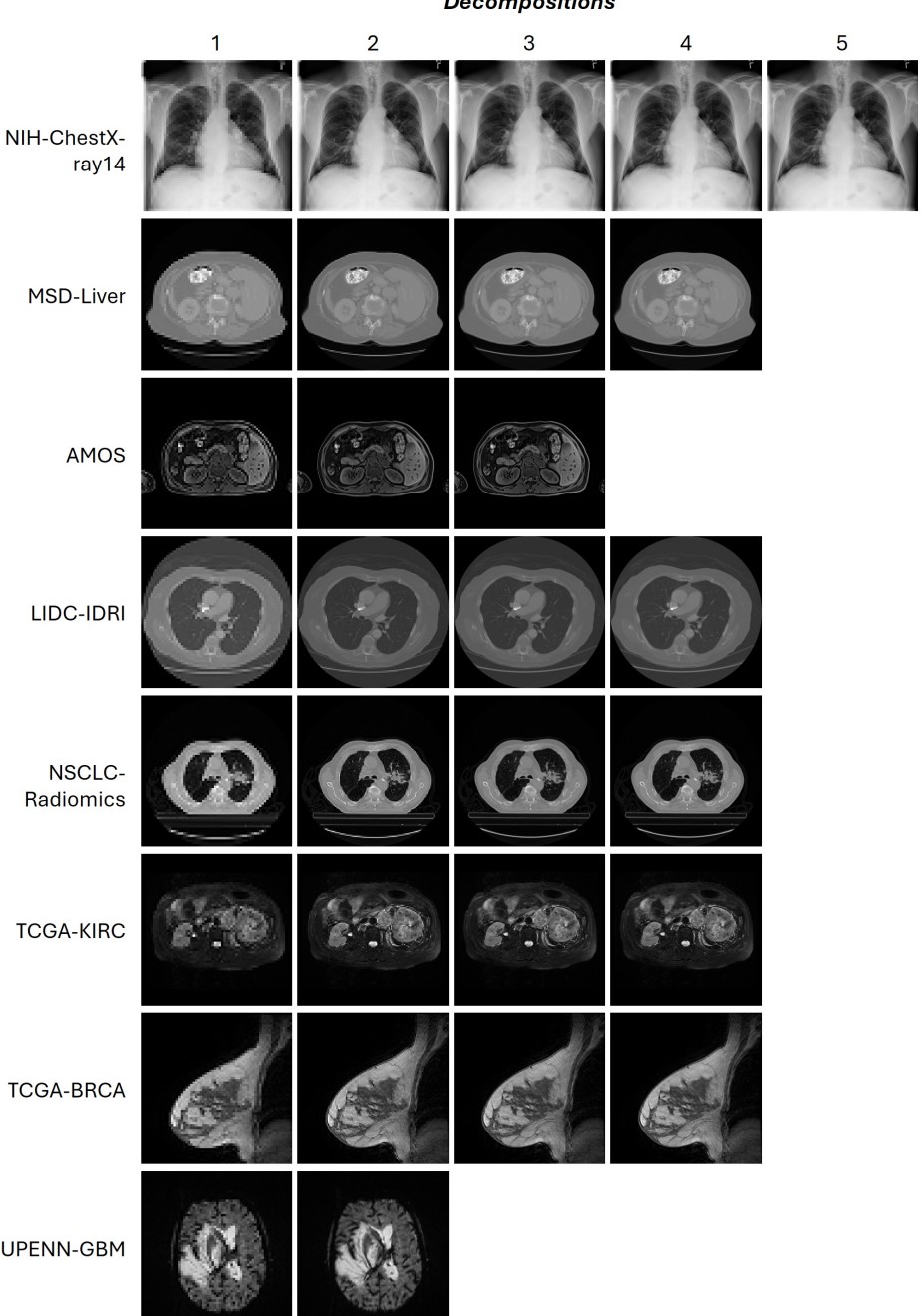

Figure D.1: Examples from MIST-encoded medical imaging datasets. The number of decompositions depends on the medical image's resolution.

