# OpenReview forum: "Towards Resource-Efficient Streaming of Large-Scale Medical Image Datasets for Deep Learning"
_MIDL.io/2025/Conference — MIDL 2025 Poster_

### Official Review · Reviewer_R1J1 · 2025-02-08

**Confidence:** 3
**Preliminary Rating:** 4
**Recommendation:** Oral

**Summary:**

**Key Ideas**

The paper introduces the Medical Image Streaming Toolkit (MIST), a novel approach designed to efficiently store and retrieve large-scale medical imaging datasets for deep learning (DL) applications.

MIST functions as a format-agnostic database, enabling researchers to access images at various resolutions and formats from a single high-resolution source. This eliminates the need for multiple dataset copies, significantly reducing both storage costs and bandwidth usage.

The study highlights challenges in medical image storage, including bandwidth limitations and preprocessing inefficiencies, and presents MIST as a solution that enables progressive image streaming on demand.

**Experiments**

The researchers evaluated MIST across eight large-scale medical imaging datasets spanning different modalities such as CT, MRI, and X-ray, and formats including DICOM, NIfTI, and PNG.

The system incorporates High-Throughput JPEG 2000 (HTJ2K) compression, allowing sub-resolution image retrieval without compromising quality.

Experimental results demonstrated a 56.22% reduction in storage requirements and up to 96% lower data transmission when accessing lower-resolution images.

Image quality was assessed using Structural Similarity Index Measure (SSIM) and Peak Signal-to-Noise Ratio (PSNR), confirming that MIST maintains high fidelity even at reduced resolutions.

**Significance**

MIST tackles the resource-intensive challenges of handling vast medical imaging datasets in DL research.

By streamlining storage and retrieval, it reduces costs, improves accessibility, and eliminates redundant preprocessing, ultimately enhancing research efficiency.

The ability to dynamically stream images rather than downloading entire dataset minimizes computational overhead, making large-scale DL more feasible in medical imaging.

This framework lays the groundwork for scalable, data-efficient medical imaging solutions and could accelerate AI-driven advancements in healthcare.

**Strengths:**

1.Practical Significance and Impact

One of the strongest aspects of this paper is its direct applicability to medical imaging and deep learning research.

The Medical Image Streaming Toolkit (MIST) provides a format-agnostic solution for storing and streaming medical images at various resolutions and formats without the need for redundant copies.

This innovation is particularly valuable as large-scale medical datasets often suffer from storage and bandwidth constraints, making deep learning research both expensive and inefficient.

By reducing storage by 56.22% and bandwidth consumption by up to 96%, MIST enhances accessibility, particularly for researchers with limited computational resources.

2. Novelty and Technical Merit

While medical image compression is a well-established field, MIST introduces a novel progressive streaming approach powered by HTJ2K compression.

This method allows researchers to access images at different resolutions on demand, minimizing unnecessary downloads. What sets MIST apart is its ability to dynamically store and stream DICOM, NIfTI, and PNG formats without requiring multiple copies, making it a highly flexible and efficient solution.

3. Rigorous Experimental Validation

The framework’s effectiveness is rigorously evaluated across eight large-scale medical imaging datasets, covering CT, MRI, and X-ray modalities.

The study systematically quantifies storage savings, bandwidth efficiency, and image quality retention using SSIM and PSNR. The results confirm that MIST achieves near-lossless compression while maintaining diagnostic quality, ensuring its practical reliability.

The use of real-world datasets further strengthens the generalizability of the findings.

**Weaknesses:**

Benchmark against existing medical imaging storage/streaming solutions like AWS HealthImaging, NCI Imaging Data Commons, or conventional DICOM PACS systems.

Validate DL model performance on streamed images at different resolutions to ensure no accuracy loss.

Provide computational cost benchmarks for real-time streaming, including latency, decoding speed, and hardware requirements.

Discuss MIST's limitations for high-resolution volumetric imaging and specialized AI models that require full-resolution data.

Address security and compliance issues related to medical data transmission and storage.

**Detailed Comments:**

The paper could explicitly outline when MIST is most beneficial (e.g., for classification tasks requiring downsampled images) and when it may not be ideal (e.g., high-resolution volumetric analysis).

A small "Limitations and Future Work" section would enhance transparency.

The paper discusses storage and bandwidth efficiency but does not address decoding time and resource consumption during streaming.
A simple runtime performance table comparing MIST vs. direct file downloads would provide a clearer picture of real-world usability.

Since MIST allows on-the-fly format conversion, it would be useful to discuss how metadata is handled when converting between formats (e.g., DICOM to PNG).

Does MIST preserve key metadata tags (e.g., patient ID, modality, imaging parameters) during format conversion?

The HTJ2K compression and progressive streaming approach is well-explained, but a visual diagram illustrating how images are progressively decomposed and streamed would be very helpful for readers unfamiliar with the method.

Table 1 lists the datasets used, but a brief paragraph summarizing key dataset differences (modality, resolution, size) before introducing the table would improve readability.

Some datasets are referred to with abbreviations (e.g., TCGA-KIRC, LIDC-IDRI) before they are fully introduced—expanding abbreviations on first mention would help clarity.


The streaming-based nature of MIST could make it useful for federated learning (FL) in medical AI, where datasets are stored across different institutions and not downloaded locally.

**Justification Of The Preliminary Rating:**

The paper presents a highly practical and well-structured solution to a pressing issue in medical imaging research: the storage and bandwidth challenges of large-scale datasets. By introducing the Medical Image Streaming Toolkit (MIST), the authors propose a novel, format-agnostic, and resource-efficient framework that allows researchers to stream images dynamically at different resolutions and formats, instead of downloading and storing redundant copies. This significantly reduces storage costs (by 56.22%) and bandwidth usage (by up to 96%), making deep learning (DL) research in medical imaging more accessible and scalable.

**Questions To Address In The Rebuttal:**

NA

**Special Issue:**

No

---

> ### Author Response · Authors · 2025-03-08
>
> Thank you for taking the time to provide valuable insights on our submission. We have addressed all concerns that were raised, and we feel confident that the manuscript has been improved in this process. Please see below the point-by-point response to the comments:
>
> **1. Benchmark against existing medical imaging storage/streaming solutions like AWS HealthImaging, NCI Imaging Data Commons, or conventional DICOM PACS systems.**
>
> Unlike MIST, existing solutions are primarily limited to DICOM, lack support for other formats like NIfTI or PNG, and do not provide access to sub-resolution images. Conventional DICOM PACS and AHI are designed for clinical use rather than research applications. While AHI uses HTJ2K compression, it is neither format-agnostic nor does it provide access to sub-resolution images. Similarly, IDC is a valuable resource for hosting medical imaging datasets but is limited to DICOM. Our method is designed to build upon frameworks like IDC by providing a flexible and scalable framework for streaming large-scale medical imaging datasets.
>
> **2. Validate DL model performance on streamed images at different resolutions to ensure no accuracy loss.**
>
> Please refer to our general rebuttal response #1.
>
> **3. Provide computational cost benchmarks for real-time streaming, including latency, decoding speed, and hardware requirements.**
>
> Please refer to our general rebuttal response #2.
>
> **4. Discuss MIST's limitations for high-resolution volumetric imaging and specialized AI models that require full-resolution data.**
>
> A primary limitation of our method for volumetric imaging is its use of slice-wise compression. Formats like NIfTI apply compression to the entire volume, whereas MIST compresses each slice individually. However, despite this limitation, our method significantly reduces storage and bandwidth required to streaming full-resolution volumes when compared to existing methods.
>
> **5. Address security and compliance issues related to medical data transmission and storage.**
>
> While security and compliance are critical for clinical deployment, our work focuses on improving data hosting and sharing for research applications, where datasets are typically anonymized and publicly accessible. A key challenge is that users may not have the storage of bandwidth to download large datasets like NLST (~11 TB). Our method addresses this challenge to make such large-scale datasets more accessible to researchers in resource-limited settings. That said, we recognize that security and compliance are relevant considerations even in research settings. This provides us with an interesting direction for future work.
>
> **6. The paper could explicitly outline when MIST is most beneficial (e.g., for classification tasks requiring downsampled images) and when it may not be ideal (e.g., high-resolution volumetric analysis).**
>
> The benefits of MIST are highly application-dependent, making it difficult to cover all possible use cases extensively. Tasks that involve downsampled images benefit the most from our method. However, even for applications requiring full-resolution images, MIST still provides significant reductions in storage and bandwidth requirements. We will revise the manuscript with these points in our discussion section.
>
> **7. A small "Limitations and Future Work" section would enhance transparency.**
>
> We will revise the manuscript to discuss the limitations and future work.
>
> **8. The paper discusses storage and bandwidth efficiency but does not address decoding time and resource consumption during streaming. A simple runtime performance table comparing MIST vs. direct file downloads would provide a clearer picture of real-world usability.**
>
> Please refer to our general rebuttal response #2.
>
> **9. Since MIST allows on-the-fly format conversion, it would be useful to discuss how metadata is handled when converting between formats (e.g., DICOM to PNG). Does MIST preserve key metadata tags (e.g., patient ID, modality, imaging parameters) during format conversion?**
>
> Currently, MIST handles the metadata by storing it in a separate JSON file. When converting between formats, the relevant metadata tags are preserved and appropriately mapped to equivalent tags in the target format (e.g., pixel spacing when converting from DICOM to NIfTI). We will revise the manuscript with this information for clarity.
>
> **10. The HTJ2K compression and progressive streaming approach is well-explained, but a visual diagram illustrating how images are progressively decomposed and streamed would be very helpful for readers unfamiliar with the method.**
>
> We agree that visualizing our methods would improve clarity. We want to point out that Figure 2(b,c) already provides a visualization of the partial streaming process. We will revise the manuscript to emphasize this figure in the methods section.

---

> > ### Author Response · Authors · 2025-03-08
> >
> > **11. Table 1 lists the datasets used, but a brief paragraph summarizing key dataset differences (modality, resolution, size) before introducing the table would improve readability.**
> >
> > We agree that a brief paragraph summarizing key dataset differences would improve readability. Due to page constraints, we opted to include Table 1 as a concise summary. However, we will add a more detailed description of datasets in the appendix.
> >
> > **12. The streaming-based nature of MIST could make it useful for federated learning (FL) in medical AI, where datasets are stored across different institutions and not downloaded locally.**
> >
> > This is an interesting point. While datasets are decentralized and not shared across institutions in FL, model weights are periodically transmitted to a central server. Exploring weight compression techniques to optimize FL communication is an interesting direction. However, it would require extensive theoretical and experimental work, and is beyond the scope of this work.

---

### Official Review · Reviewer_mC2u · 2025-02-19

**Confidence:** 4
**Preliminary Rating:** 4
**Recommendation:** Poster
**Final Rating:** 4

**Summary:**

The paper introduces the Medical Image Streaming Toolkit (MIST), a format-agnostic framework designed to address the ever-growing challenges of hosting and accessing large-scale medical imaging datasets for deep learning research. By leveraging progressive encoding via HTJ2K, MIST enables researchers to stream medical images at various resolutions and formats directly from a single high-resolution copy. The authors evaluate their approach on eight diverse datasets—ranging from CT and MR to PNG and DICOM formats—demonstrating significant reductions in both storage (up to ~71%) and data transmission (up to ~98% at lower resolutions) while preserving image quality as measured by SSIM and PSNR. The comprehensive experimental setup and quantitative analysis underscore the potential of MIST to not only reduce infrastructure overhead but also streamline data preprocessing pipelines for DL applications in medical imaging. Overall, the paper contributes a novel and practically relevant solution that could substantially lower the barrier for DL researchers by enabling flexible, resource-efficient data access.

**Strengths:**

1. MIST’s design addresses a critical need in medical imaging research by allowing a single high-resolution copy to be used for multiple downstream tasks. This format-agnostic approach is particularly compelling given the diversity of imaging data and use cases in the field.
2. The authors provide a detailed quantitative analysis across eight popular datasets. The experiments clearly illustrate substantial savings in storage and bandwidth, which are supported by rigorous metrics (SSIM, PSNR) and comprehensive tables.
3. With publicly available code on GitHub and a clear methodology, the work is reproducible and has immediate applicability in reducing preprocessing time and resource costs. The ability to seamlessly convert between formats (e.g., DICOM to NIfTI) is a strong asset for the community.
4. The use of HTJ2K to support sub-resolution streaming is well motivated. This not only minimizes data transfer but also preserves the necessary image quality for deep learning applications, making it a pragmatic solution for resource-constrained environments.

**Weaknesses:**

1. Although the paper provides quantitative metrics for data reduction and image quality, it stops short of evaluating the impact of streaming sub-resolution images on downstream DL model performance. While previous work is referenced, including additional end-to-end experiments (e.g., segmentation or classification accuracy) would strengthen the claims regarding the negligible impact on DL tasks.
2. The paper does not fully address potential latency issues or integration challenges when incorporating MIST into existing DL pipelines. For instance, the real-time performance and any overhead introduced by on-the-fly format conversions are not discussed, which could be critical for certain clinical applications.
4. Although the format-agnostic approach is a strength, the paper could benefit from a more detailed discussion of potential pitfalls related to metadata conversion—especially when dealing with complex DICOM metadata—and how these issues might affect clinical validity or interoperability.
5. While the experimental results are promising, it is unclear how the framework scales when handling images with extreme variability in resolution and bit depth. A more granular analysis at the individual image level, rather than aggregated metrics, would provide deeper insights into performance variations across different data subsets.

**Detailed Comments:**

- The experimental design is robust, yet more insights into how different decomposition levels affect model performance in practice would be valuable.
- The authors mention minor data exclusions (e.g., a small percentage of series that failed to encode) but could elaborate on whether these exclusions might introduce any bias in downstream tasks.
- It would be beneficial to include a discussion on the potential for real-time streaming in clinical settings, where latency and data security are paramount.

**Justification Of The Final Rating:**

Thank you to the authors for the detailed response. I will maintain my rating at 4, as potential pitfalls related to metadata conversion are designated as limitations and for future work rather than being addressed in the current study.

**Justification Of The Preliminary Rating:**

The paper presents a significant contribution by proposing an innovative and practical solution for the streaming of large-scale medical imaging data. Its strengths lie in the novel format-agnostic design, substantial storage and bandwidth savings, and thorough quantitative evaluation. However, the work would be further strengthened by a more comprehensive evaluation of the impact on downstream deep learning tasks and a deeper discussion of integration challenges in real-world applications. Despite these minor shortcomings, the potential for reducing infrastructural overhead and streamlining data preprocessing for DL models makes this work a valuable addition to the field.

**Questions To Address In The Rebuttal:**

1. Could the authors provide additional experimental results or case studies demonstrating the impact (or lack thereof) of using sub-resolution images on the performance of deep learning models in actual clinical tasks?
2. How does MIST handle the challenges of latency and potential integration bottlenecks when deployed in real-world DL pipelines, particularly in resource-constrained or real-time environments?
3. Can the authors discuss any observed limitations regarding metadata fidelity during format conversion and the potential implications for clinical interpretations?

**Special Issue:**

No

---

> ### Author Response · Authors · 2025-03-08
>
> Thank you for taking the time to provide valuable insights on our submission. We have addressed all concerns that were raised, and we feel confident that the manuscript has been improved in this process. Please see below the point-by-point response to the comments:
>
> **1. Could the authors provide additional experimental results or case studies demonstrating the impact (or lack thereof) of using sub-resolution images on the performance of deep learning models in actual clinical tasks?**
>
> Please refer to our general rebuttal response #1.
>
> **2. How does MIST handle the challenges of latency and potential integration bottlenecks when deployed in real-world DL pipelines, particularly in resource-constrained or real-time environments?**
>
> Please refer to our general rebuttal response #2.
>
> **3. Although the format-agnostic approach is a strength, the paper could benefit from a more detailed discussion of potential pitfalls related to metadata conversion—especially when dealing with complex DICOM metadata—and how these issues might affect clinical validity or interoperability.**
>
> We agree that further evaluation is required to identify the best approach of handling complex DICOM metadata. In our experiments, we used JSON to store and transmit DICOM metadata, but we recognize that this approach may not scale efficiently when the metadata is complex. One potential solution would be to transmit the original DICOM header. This would be an interesting direction for future work and we will include it as both a limitation and a future research direction in our discussion section.
>
> **4. While the experimental results are promising, it is unclear how the framework scales when handling images with extreme variability in resolution and bit depth. A more granular analysis at the individual image level, rather than aggregated metrics, would provide deeper insights into performance variations across different data subsets.**
>
> We agree that a more granular analysis would provide deep insights into MIST’s scalability. While we had previously acknowledged this as a limitation in our discussion section, we will expand our results by incorporating image quality metrics stratified by the maximum decomposition level of an image, representing variation in image resolution.
>
> **5. The authors mention minor data exclusions (e.g., a small percentage of series that failed to encode) but could elaborate on whether these exclusions might introduce any bias in downstream tasks.**
>
> Please refer to our general rebuttal response #3.

---

### Official Review · Reviewer_RwLB · 2025-02-22

**Confidence:** 3
**Preliminary Rating:** 2
**Final Rating:** 3

**Summary:**

- The manuscript proposes a format-agnostic database (named MIST) to stream medical images at different resolutions and formats from a single high-resolution copy. Given the large size of medical images, scalable deep learning infrastructure is crucial but remains underexplored. The work addresses the gap.
- The proposed method uses HTJ2K, a progressive encoding format, allowing extracting low-resolution image directly from the compressed codestream without fully decoding images.
- The experiments include 8 large-scale datasets. The authors encoded and stored all images, measured the data storage and transmission size, and image quality at different decompositions.

**Strengths:**

- The manuscript explores how to build infrastructure for transmitting large-scale medical images for deep learning, which is an essential but underexplored topic.
- The writing is clear and easy to follow.

**Weaknesses:**

- Though the manuscript addresses an underexplored but essential topic, the technical innovation is limited. The proposed method (mainly using HTJ2K to effectively stream low-resolution images) is straightforward.
- The proposed system has not been tested under actual deep learning training. Deep learning training demands high-speed data transmission with minimal latency. To ensure this requirement is met using a standard internet connection—while avoiding the need for research institutions to store large volumes of data—a well-designed buffering system is essential. How can such a system be structured to achieve optimal performance?
- Certain medical images, such as CT scans, require different windowing settings to visualize various tissues. This should be considered in the design process.
- Authors may consider system designs more tailored for some common scenarios. For example, a lot of models are trained on image patches (e.g. denoising models, some segmentation models). How can the system be optimized for this kind of tasks?

**Detailed Comments:**

- In "3.2 Experiments": "Any series that failed to encode due to unsupported pixel data was excluded from our analysis." What kind of pixel data were unsupported?

**Justification Of The Final Rating:**

Parts of my concerns were addressed during the rebuttal. The additional deep learning experiment is a plus. I have a better understanding of the work's motivation. But my opinion on the technical innovation remains unchanged.

**Justification Of The Preliminary Rating:**

The manuscript proposed a system design to effectively stream large medical image datasets for deep learning training, avoiding the need for users to store these data. The topic is underexplored. However, not much technical innovation is presented and some necessary factors to consider when building such systems are missing.

**Questions To Address In The Rebuttal:**

Please see "Weaknesses" and "Detailed Comments".

---

> ### Author Response · Authors · 2025-03-08
>
> Thank you for taking the time to provide valuable insights on our submission. We have addressed all concerns that were raised, and we feel confident that the manuscript has been improved in this process. Please see below the point-by-point response to the comments:
>
> **1. Though the manuscript addresses an underexplored but essential topic, the technical innovation is limited. The proposed method (mainly using HTJ2K to effectively stream low-resolution images) is straightforward.**
>
> While our method is straightforward, its novelty lies in its ability to encode medical images from various formats into a single, format-agnostic representation – whereas current solutions are limited to DICOM. Unlike existing methods, our approach not only reduces storage and bandwidth requirements, but also enables access to sub-resolution images. By integrating these components into a unified framework, our method supports the diverse needs of researchers using public, large-scale medical imaging datasets. We believe that this flexibility and scalability is a meaningful contribution to the MIDL community.
>
> **2. The proposed system has not been tested under actual deep learning training. Deep learning training demands high-speed data transmission with minimal latency. To ensure this requirement is met using a standard internet connection—while avoiding the need for research institutions to store large volumes of data—a well-designed buffering system is essential. How can such a system be structured to achieve optimal performance?**
>
> Please refer to our general rebuttal responses #1 and #2.
>
> **3. Certain medical images, such as CT scans, require different windowing settings to visualize various tissues. This should be considered in the design process.**
>
> Our method preserves pixel data using scaling, ensuring no loss of precision. As a result, when decoded, the pixel data can be windowed at different levels without any loss of information. The main concern arises when a DICOM image (typically with 16-bit pixel depth) is converted into a format like JPEG (8-bit pixel depth), leading to potential loss of information. However, in our method, the underlying hosted pixel data remains unchanged from the original DICOM format. Additional edge cases are discussed in general rebuttal response #3.
>
> **4. Authors may consider system designs more tailored for some common scenarios. For example, a lot of models are trained on image patches (e.g. denoising models, some segmentation models). How can the system be optimized for this kind of tasks?**
>
> For scenarios where models are trained on image patches, a researcher could still benefit from downloading the full-resolution data, as MIST significantly reduces storage and bandwidth requirements compared to existing methods. While our goal is to provide a flexible framework that supports diverse research use-cases rather than optimizing for a single scenario, our approach could indeed be adapted to optimize streaming of patches. Exploring such techniques could be an interesting direction for future work. Additionally, we used sub-resolution images to train a patch-based segmentation model in general rebuttal response #1.
>
> **5. In "3.2 Experiments": "Any series that failed to encode due to unsupported pixel data was excluded from our analysis." What kind of pixel data were unsupported?**
>
> Please refer to our general rebuttal response #3.

---

> > ### Comment · Reviewer_RwLB · 2025-03-12
> >
> > I would like to thank the authors for their feedback. Parts of my concerns are addressed. But my opinion on the technical innovation remains unchanged. Therefore, I raised my rating to 3.

---

### Author Rebuttal · Authors · 2025-03-08

We thank the reviewers for their valuable feedback. In this general rebuttal, we aim to address the common concerns raised by the reviewers:

**1. Deep Learning Experiments**

We conducted two preliminary experiments to validate MIST-streamed sub-resolution images for training DL models. In both experiments, we trained models on both the original and sub-resolution images, then tested them on the original internal and external datasets.

_2D Chest X-Ray Classification_

We randomly split the NIH-ChestX-ray14 dataset on the patient-level into train ($70\\%$, $n=78,075$), validation ($10\\%$, $n=11,079$), and test ($20\\%$, $n=22,966$) sets. We used $n=15,000$ randomly sampled frontal chest x-rays from the MIMIC-CXR dataset as our external test set. Then, we trained two ImageNet-pretrained DenseNet121 models at resolution $224\times224$, one on the original PNG images and the other on the sub-resolution images. We measured the performance of both models using the mean AUROC scores of seven labels: Atelectasis, Cardiomegaly, Consolidation, Edema, Pleural effusion, Pneumonia, and Pneumothorax.

On the NIH-ChestX-ray14 test set, we observed that both the PNG-trained and MIST-trained models measured a mean AUROC of $0.84 \pm 0.05$. When evaluated on the external MIMIC-CXR test set, both models similarly measured a mean AUROC of $0.76 \pm 0.07$.

_3D Liver Segmentation_

We randomly split the MSD-Liver dataset into train ($80\\%$, $n=106$) and test ($20\\%$, $n=25$) sets. We used $n=30$ volumes from the BTCV dataset  as our external test set. Then, we trained two 3D-UNet models at resolution $256\times256\times128$, voxel spacing $1.5\times1.5\times2.0$, and random foreground patches of resolution $128\times128\times32$. We measured the performance of both models using the mean Dice score.

On the MSD-Liver test set, we observed that the NIfTI-trained model measured a mean Dice score of $0.95 \pm 0.02$, while the MIST-trained model measured a mean Dice score of $0.94 \pm 0.02$. When evaluated on the external BTCV test set, both models measured a mean Dice score of $0.93 \pm 0.02$.

While our work primarily focuses on building a framework for data hosting and sharing in research settings, our previous work [1] explored the inference performance of sub-resolution images in a clinical deployment setting, but did not involve training models with such images. We will revise the manuscript to include these preliminary DL experiments.

---

> ### Author Response · Authors · 2025-03-08
>
> **2. Concerns of Latency and Challenges in Integration**
>
> The primary goal of our paper is to build a flexible and scalable framework for hosting and sharing medical imaging datasets in research settings, with a focus on reducing data storage and transmission. Our approach is not intended for real-time DL applications, but rather for efficient data access across diverse research settings, where datasets are typically downloaded before being used for DL tasks. For example, a researcher downloading the NLST dataset (~11 TB) could benefit from our method, as the computational cost of format conversion is significantly lower than the storage and bandwidth required to download the dataset on workstation-level hardware.
>
> While our method addresses challenges related to data storage and transmission for research, it is not intended for clinical deployment. However, in our previous work [1], we explored real-time streaming in clinical settings, where latency is critical, and demonstrated significant improvements in key metrics such as decoding time, inference time, and model throughput. We will revise the manuscript to make this distinction clear.
>
> **3. Clarification of Exclusion**
>
> In our analysis, we excluded images that contained unsupported pixel data. Specifically, our current implementation of HTJ2K supports up to 16-bit precision, and any image exceeding this limit would result in either loss of information or clipping. For example, certain MRI volumes in AMOS exceeded the 16-bit pixel depth, potentially due to the presence of artifacts, and were excluded. Additionally, we excluded DICOM files that did not contain pixel data, such as SEG files. We acknowledge that these exclusions could potentially bias downstream tasks, especially if the excluded data is correlated with specific image characteristics. We plan to explore this further in future work and will revise the manuscript to include this point as a limitation in our discussion section.
>
> **References**
>
> [1] Kulkarni, P., Kanhere, A., Siegel, E. L., Yi, P. H., & Parekh, V. S. (2024). ISLE: An Intelligent Streaming Framework for High-Throughput AI Inference in Medical Imaging. Journal of Imaging Informatics in Medicine, 37(6), 3250-3263.

---

### Meta-Review · Area_Chair_RQbF · 2025-03-21

**Recommendation:** Accept (Poster)
**Confidence:** 4

**Metareview:**

This paper presents a tool to compress medical imaging databases. The authors propose a format-agnostic framework to compress medical images from different body parts and modalities. While there are not many technical contributions in this paper, I believe it is a useful tool. The deep learning experiments added during the rebuttal period further demonstrate the utility of MIST.

In my opinion, this is a borderline paper, showing clear advantages (a new tool to solve a practical problem) and weaknesses (limited novelty). Personally, I am inclined to accept it because I appreciate the authors' efforts in introducing a practical tool. I suggest that the authors validate MIST with more deep learning applications, conduct stress tests with MIST, and report the curve of 'memory saving' versus 'performance drop' in the next version of their technical report.